# A Systematic Review and Meta-Analysis of Fracture-Related Infections in Maxillofacial Trauma: Incidence, Risk Factors, and Management Strategies

**DOI:** 10.3390/jcm14041332

**Published:** 2025-02-17

**Authors:** Frederic Van der Cruyssen, Millie Forrest, Simon Holmes, Nabeel Bhatti

**Affiliations:** 1Department of Oral and Maxillofacial Surgery, University Hospitals Leuven, 3000 Leuven, Belgium; 2OMFS-IMPATH Research Group, KU Leuven, 3000 Leuven, Belgium; 3Department of Oral and Maxillofacial Surgery, Royal London Hospital, London E1 1BB, UK

**Keywords:** fracture-related infections, maxillofacial trauma, meta-analysis, systematic review

## Abstract

**Background/Objectives**: Fracture-related infections (FRIs) are a significant complication in maxillofacial trauma, leading to adverse outcomes such as prolonged healing, nonunion, and osteomyelitis. Despite advancements in surgical techniques, the incidence of FRIs remains concerning, particularly for mandibular fractures. This systematic review and meta-analysis aims to evaluate the incidence, risk factors, and management strategies for FRIs in oral and maxillofacial trauma. **Methods**: A systematic search of Medline and Embase databases was conducted, including studies up to February 2024, adhering to PRISMA guidelines. Eligible studies included randomized controlled trials, cohort studies, and case-control studies focusing on the incidence, risk factors, or treatment outcomes of FRIs. Data on patient demographics, fracture type, infection rates, and management strategies were extracted and analyzed. Statistical analyses included pooled infection rates, stratified by anatomical sites, using fixed and random-effects models. **Results**: A total of 72 studies were included, with a pooled FRI rate of 5.6%. Mandibular fractures exhibited the highest infection rate at 8.9%, while midface fractures had the lowest at 0.9%. The significant risk factors identified included smoking, substance abuse, and comorbidities such as diabetes. Delayed surgical intervention and poor periodontal health were also associated with higher infection rates. The use of prophylactic antibiotics did not show significant differences in infection prevention. **Conclusions**: FRIs in maxillofacial trauma are influenced by multifactorial risks, including anatomical, patient-specific, and procedural factors. Mandibular fractures are particularly vulnerable, requiring targeted preventive strategies and timely intervention. Standardized definitions and guidelines are essential for improving consistency in diagnosis and management. Future research should focus on high-quality trials to establish evidence-based approaches for FRI prevention and treatment.

## 1. Introduction

Maxillofacial fractures are a common outcome of traumatic injuries, with the mandible being the most frequently affected bone in the facial skeleton [1,2,3]. These fractures often require surgical intervention to restore function and aesthetics. Despite advancements in surgical techniques and postoperative care, fracture-related infections (FRIs) remain a significant complication in the management of these injuries [4]. Such infections can lead to prolonged hospitalization, increased morbidity, and adverse clinical outcomes, including nonunion of fractures and the development of chronic osteomyelitis [4].

Fracture-related infection (FRI) is defined as an infection associated with a bone fracture, which may occur with or without operative intervention [5]. Unlike other musculoskeletal infections, FRIs are characterized by biomechanical instability, frequent involvement of soft tissue, and potential vascular damage, all of which complicate the healing process and increase the infection risk. This entity differs from other forms of infections such as acute osteomyelitis or odontogenic infections, emphasizing the specific challenges associated with trauma, particularly the emergency nature of many cases that prevent preoperative optimization [5].

The definition and interpretation of fracture-related infections encompass a broad spectrum, ranging from minor issues such as plate exposure without significant clinical symptoms to more severe cases involving nonunion with osteomyelitis and abscess formation. This variability reflects the complexity and challenges inherent in managing FRIs, as each case may present differently based on the severity of the infection, the anatomical site, and patient-specific factors. Understanding this spectrum is crucial for guiding clinical decision-making, as the treatment approach must be individualized to the specific presentation of FRI.

The incidence of FRIs in oral and maxillofacial surgery varies widely across the literature, with reported rates up to 20%, depending on factors such as patient demographics, fracture type, and surgical approach [3]. Several risk factors have been implicated in the development of postoperative infections [5]. Patient-related factors include smoking, substance abuse, and underlying comorbidities like diabetes mellitus [6,7]. Surgical factors encompass the timing of intervention, the complexity of the fracture, and the choice of fixation method. Additionally, perioperative management strategies, particularly the use of prophylactic antibiotics, play a crucial role in infection prevention [8].

Understanding the multifactorial nature of FRIs is essential for improving patient outcomes. While individual studies have explored specific aspects of infection risk and management, a comprehensive synthesis of the available evidence is lacking. Therefore, the objective of this systematic review is to evaluate the incidence of fracture-related infections in maxillofacial trauma, identify key risk factors contributing to postoperative infections, and assess the effectiveness of various management strategies. By consolidating current knowledge, this review aims to provide clinicians with evidence-based guidance for the prevention and management of infections in patients undergoing surgery for maxillofacial fractures.

## 2. Materials and Methods

### 2.1. Search Strategy

This study was registered in PROSPERO with identifier CRD42024512015 and adheres to the PRISMA statement for systematic reviews (PRISMA checklist Appendix A) [9]. A comprehensive search of Medline and Embase databases was conducted using a combination of keywords and MeSH terms related to oral and maxillofacial surgery, bone fractures, and infections (Table 1).

The search included studies published up to February 2024. The search terms included variations such as “oral surgery”, “maxillofacial surgery”, “fracture”, and “fracture-related infection”. No restrictions were placed on language, but the review focused on human studies only.

### 2.2. Inclusion and Exclusion Criteria

Eligible studies included randomized controlled trials (RCTs), cohort studies, case-control studies, and cross-sectional studies. Studies were included if they:Involved patients undergoing oral and maxillofacial surgery for fractures.Reported on the incidence, risk factors, diagnostic methods, or treatment outcomes of fracture-related infections.

Studies were excluded if they:Were case reports, reviews, systematic reviews, meta-analyses, surveys, editorials, or opinion pieces.Focused on fractures unrelated to the oral and maxillofacial region, or infections unrelated to fracture management.

### 2.3. Data Extraction and Risk of Bias Assessment

Two independent and blinded reviewers (MF, FVDC) extracted data from each study, including details on patient demographics, fracture type, surgical technique, infection incidence, risk factors, and treatment outcomes using Rayyan (Rayyan Systems Inc., Cambridge, MA, USA) [10]. Study quality was assessed using the Newcastle Ottawa Scale, and a subsequent AHRQ (Agency for Healthcare Research and Quality) composite score was constructed [11]. The overall quality of the studies was classified using the following composite scores: “Good”, “Fair”, or “Poor”. Disagreements were resolved through discussion, and a third reviewer was consulted if necessary. Only studies that received a “Fair” or “Good” score were assessed for risk factors and management strategies.

### 2.4. Statistical Analysis

The infection rates were calculated by extracting data from the included studies and aggregating the number of reported infections relative to the total sample size and further stratified for the anatomical site (mandible, condyle, midface, or frontal) if possible. A single-arm meta-analysis was conducted to estimate the pooled infection rate. The analysis utilized both fixed-effects and random-effects models to account for between-study heterogeneity, with heterogeneity quantified using I^2^ statistics. Proportions and corresponding 95% confidence intervals were calculated for each study. Statistical analyses were performed using R version 4.3.3 (R Core Team, Vienna, Austria). We further stratified infection rates according to the year of publication to assess the trends of infection rates over the years. Finally, we examined whether studies reporting the use of prophylactic antibiotics in fracture management exhibited different infection rates compared to those that did not, using a two-sided *t*-test.

## 3. Results

A total of 963 records were identified through database searching. After removing duplicates, these records were screened based on title and abstract, and 178 full-text articles were assessed for eligibility. Ultimately, 72 studies were included in the final review (Appendix A). The PRISMA flowchart detailing the study selection process is provided in Figure 1.

The risk of bias assessment across the studies revealed variability in methodological quality, as indicated by the AHRQ Quality Rating (Figure 2, Appendix A). The majority of the studies received an AHRQ “Good” or “Fair” rating, reflecting acceptable methodological rigor.

Several studies exhibited shortcomings in areas such as prospective data collection, unbiased assessment, and adequacy of control groups.

The calculated I^2^ statistic was 92.6%. Due to significant heterogeneity, we did not attempt a meta-analysis for risk factors and management strategies.

### 3.1. Infection Rate

The overall pooled infection rate for maxillofacial fractures was 5.6% (Figure 3).

Specifically, mandibular fractures had an infection rate of 8.9%, midface fractures (including zygomatic and maxillary) had a rate of 0.9%, condylar fractures exhibited a rate of 1.8%, and frontal bone fractures had a 2.7% infection rate (Figure 4).

A review of reported infection rates over the years revealed no significant differences (Figure 5). Similarly, no notable variation in infection rates was observed between studies that reported the use of prophylactic antibiotics and those that did not (7.1% vs. 6.4%, t = 0.14, *p* = 0.89).

### 3.2. Risk Factors for Fracture-Related Infections

#### 3.2.1. Smoking

Smoking was consistently identified as a significant risk factor for infections following oral and maxillofacial surgery across multiple studies. Zrounba et al. found a statistically significant higher complication rate among smokers (*p* = 0.006), with 44.4% of the delayed treatment group being smokers and all complications occurring in this group [12]. Serena-Gómez and Passeri reported that 35.3% of their patients were smokers, who had a 14.9% complication rate, primarily infections, compared to 8.5% in non-smokers [13]. Smokers in their study showed a statistically significant increase in complications compared to non-smokers (*p* < 0.05). Açil et al. also found that 77.8% of patients with osteosynthesis-related infections were smokers, with a longer smoking history significantly associated with infection (*p* = 0.010) [14].

#### 3.2.2. Substance Abuse

In a study of 120 patients, 29.2% had histories of substance abuse (including alcohol abuse, intravenous drug use, and HIV), with a significantly higher postoperative infection rate of 22.9% among this group compared to 10.5% in patients without such histories [15]. Alcohol abuse and intravenous drug use were particularly associated with higher infection rates, at 28.6% and 22%, respectively [15]. Another study of 273 patients found that smoking (83.3%), alcohol use (77.8%), and drug use (50%) were significantly correlated with long-term complications like infection and hardware exposure [16]. Similarly, a larger study of 1399 patients with mandibular fractures reported that intravenous drug users had the highest complication rate (37.5%), followed by alcohol abusers (17.1%) and smokers (14.9%) [13]. In contrast, patients without substance abuse had a lower complication rate of 8.5%. Across these studies, substance abuse—particularly smoking, alcohol, and drug use—was consistently linked to higher rates of infections, malunion, and other postoperative complications.

#### 3.2.3. Age

The influence of age as a risk factor for FRIs was assessed across five studies. In a study detailed by Kerdoud et al., 196 patients were categorized into three age groups with mean ages of 33, 43, and 47 years [17]. The infection rates among these groups were reported, showing no statistically significant differences (*p* > 0.05). Conversely, a review by Jazayeri et al. showed that increasing age (odds ratio [OR], 1.03; 95% confidence interval [CI], 1.03 to 1.04; *p* < 0.0001) was an independent risk factor for the development of any adverse event. It also showed that increasing age (OR, 1.05; 95% CI, 1.04 to 1.06; *p* < 0.0001) and obesity (OR, 1.77; 95% CI, 1.05 to 2.98; *p* = 0.033) were independent risk factors for the development of a serious adverse event [18]. In pediatric populations, Bansal et al. reported no FRIs in a comparison of ORIF and closed treatment among children [19]. Hsueh et al. reported that younger adults in their study of mandibular fractures experienced fewer complications. However, these patients were more likely to be treated with a closed approach, although exact age stratifications and corresponding infection rates were not detailed, the trend suggested a lower risk associated with younger ages. Pitak-Arnnop et al. identified a significant increase in surgical site infections among older (age ≥ 60 years) COVID-19 patients, with a relative risk of 1.56 (r = 0.49; *p* = 0.0001) for developing infections, emphasizing that age and systemic health conditions combined significantly elevate infection risks [20].

#### 3.2.4. Surgical Techniques

Three studies were identified discussing FRIs in relation to surgical technique. Intraoral surgical approaches were significantly associated with increased rates of infection compared to extraoral approaches (*p* = 0.004) in a study by Açil et al., 2017 [14]. Another study showed that intraoral access resulted in a higher frequency of complications, with five out of nine patients (55.56%) experiencing issues, including infections, compared to two out of nine patients (22.22%) for extraoral access. The overall infection rate in this study was 22.22%, with intraoral approaches showing higher complication rates, although this was not statistically significant (*p* = 0.564) [21]. James et al. found no statistically significant difference in infection rates between different surgical approaches (*p* = 0.203) [22].

#### 3.2.5. Fixation Method

The studies reviewed focused on patients with mandibular fractures, predominantly involving young to middle-aged adults, with the majority being male. The types of fractures varied between mandibular angle and symphyseal regions. In Kerdoud et al., 112 patients with mandibular angle fractures were treated using single or double miniplates, and infection occurred in 36 patients, but there was no statistically significant difference between the groups [17]. Schierle et al. studied 31 patients with mandibular angle fractures treated with either one or two miniplates and found no significant differences in infection rates between the two fixation methods [23]. Similarly, Ferreira E. Costa et al. reviewed 19 patients treated with one of three fixation methods for isolated mandibular angle fractures, and again, there were no significant differences in infection rates between the groups [21]. Lastly, Agnihotri et al. compared 80 patients with mandibular symphyseal fractures treated using either miniplates or cortical screws and reported only one postoperative infection, which was not significantly different between the groups [24]. The comparison of fixation types as a risk factor for infection could not be identified for other anatomical areas beyond the mandibular angle and symphyseal regions based on the selected papers.

#### 3.2.6. Comorbidities

In a study examining mandibular fractures, the postoperative infection rate was significantly higher among patients with comorbid conditions, including diabetes, HIV, alcohol abuse, or intravenous drug abuse, with a rate of 22.9% compared to 10.5% among patients without these comorbidities [15]. Similarly, in a study on patients with asymptomatic or mildly symptomatic SARS-CoV-2 infection, patients showed a significantly higher infection rate compared to the control group (19.1% vs. 6.8%, *p* < 0.0001), with an odds ratio of 3.22 (95% CI 2.17 to 4.78) [20]. Surprisingly, no studies investigated ASA status as a risk factor for FRIs or postoperative complications.

#### 3.2.7. Time Delay

In a study by James et al., a cohort of 64 patients treated for mandibular fractures showed a mean delay to surgery of 17.52 days among those who developed complications, compared to 13.26 days for those without complications. However, the difference was not statistically significant for infection incidence (*p* = 0.203) or other complications, such as cranial nerve injury and mechanical failure [22]. Similarly, Malanchuk and Kopchak reported that delayed treatment of more than seven days was associated with a significant increase in the infection rate, rising from 4.3% for those treated on the first day to 55% among those treated after seven days [7]. Zrounba et al.’s study on mandibular fractures indicated that a significant relationship between the delay to surgery and infection could not be confirmed (*p* = 0.994) [12]. The comparative studies of treatment modalities by Agnihotri et al. and Anslem et al. both highlighted that overall postoperative complication rates, including infections, were generally higher when treatment was delayed, but numbers were too low to reach statistical significance [24,25].

#### 3.2.8. Antibiotics

In our analysis, antibiotic use in the management of mandibular fractures was heterogeneous across studies. The most commonly reported antibiotics included first-generation cephalosporins, amoxicillinclavulanic acid, metronidazole, and penicillins. Four studies included more granular data on FRIs and the effect of antibiotic use. In a study involving pediatric facial fractures, the infection rate among patients who received antibiotics was 0.9% compared to 0.5% in those who did not (*p* = 0.12), yielding an adjusted odds ratio of 1.1 (95% CI 0.53–2.34) [26]. The authors did not mention which antibiotics were used. A randomized controlled trial on mandibular fractures showed no significant difference in infection rates between a 1-day antibiotic regimen (12%) and a 5-day regimen (16%) of amoxicillinclavulanic acid with metronidazole [27]. Conversely, a case–control study of craniomaxillofacial fractures in asymptomatic or mildly symptomatic COVID-19 patients found that prolonged antibiotic use, without specifying the type of antibiotics used, significantly increased the risk of surgical site infection, with a relative risk (RR) of 2.03 (*p* = 0.0001) [20]. A separate retrospective analysis of traumatic mandibular fractures also showed no statistically significant difference in postoperative complications, including infection, based on the duration of antibiotic therapy [28]. In this study, there was a notable variation in the class of antibiotics used, which did not allow for a meaningful comparative analysis. The main classes used were amoxicillinclavulanate and macrolides.

#### 3.2.9. Preoperative Oral and Periodontal Health

One study observed a significant association between periodontal health and the incidence of osteosynthesis material (OM)-related infections [14]. The incidence of infection was significantly higher in the periodontitis group (50%, *n* = 7) compared to the healthy group (11.1%, *n* = 2) (*p* < 0.001). A positive correlation was found between periodontal disease and signs of OM-related infection (*p* = 0.022). In total, 77% of patients with clinical signs of infection had periodontal disease. The odds of developing an OM-related infection were higher in patients with a history of smoking, although no statistically significant relationship was observed between smoking and infection (*p* > 0.05). Microbiological analyses demonstrated greater microbial colonization on osteosynthesis plates from periodontitis patients, with intraorally placed plates showing a wider range of bacterial colonization compared to extraoral plates (*p* = 0.004). Intraorally placed plates were more frequently colonized by streptococci, while propionibacteria and staphylococci were less commonly detected compared to extraoral plates (*p* < 0.001 and *p* = 0.014, respectively).

### 3.3. Management Strategies

The management of FRIs across the studies involved different approaches depending on the type of fractures and fixation methods used. In a study on the use of cortical screws and miniplates for mandibular symphyseal fractures, the occurrence of postoperative infections was documented in 13 out of 80 patients, who were treated effectively through local debridement and continued care, demonstrating primary stability in the majority of cases treated with either method [24]. In a comparative analysis of ceftriaxone versus penicillin for antibiotic prophylaxis of compound mandibular fractures, both regimens showed similar efficacy, with only two infections reported per group, which resolved with local wound care, removal of the internal fixation, and oral antibiotics [29]. In a retrospective review of mandibular angle fractures, postoperative infections were effectively managed with localized incision, drainage, and antibiotic therapy, and none required further surgical intervention [30]. In one study in which zygomaticomaxillary complex (ZMC) fractures were managed using open reduction and rigid internal fixation with titanium mini plates, infections were rare, with all cases successfully resolved using standard antibiotic therapy [31]. One paper could be identified that specifically discussed the treatment of unstable, oblique, infected mandibular fractures, applying a 2.3 mm reconstruction bone plate, with no evidence of infection in follow-ups [32].

## 4. Discussion

This systematic review and meta-analysis offers the first comprehensive evaluation of the incidence, risk factors, and management strategies for FRIs in patients undergoing oral and maxillofacial surgery. Our findings highlight the multifactorial nature of FRIs and emphasize the interplay of anatomical, surgical, and patient-specific factors in influencing postoperative outcomes.

While the term “FRI” has been increasingly recognized, it was only recently introduced and remains broadly interpreted across the literature [5]. This variability in definition was reflected in many of the included studies, in which it was often unclear what specific type of infection was observed. The absence of a standardized framework for defining and categorizing FRIs introduces significant heterogeneity, complicating the comparability of findings and the establishment of evidence-based guidelines. FRIs encompass a broad spectrum of clinical presentations, ranging from asymptomatic cases with some plate exposure to severe, life-threatening infections. A key observation in our review was the variability in how studies defined or did not define reported infections, highlighting the need for greater alignment in terminology. Based on the literature, the spectrum of FRIs can be categorized as indicated in Table 2.

This proposed classification provides a framework for future research and clinical reporting to ensure greater consistency in FRI diagnosis and characterization. Standardized terminology can improve comparability between studies, facilitate meta-analyses, and guide tailored management strategies.

The overall pooled infection rate of 5.6% in this meta-analysis aligns with the variability reported in previous studies, in which infection rates range from 1% to 20%, depending on the anatomical site and patient demographics. Notably, mandibular fractures exhibited the highest infection rate at 8.9%. This higher rate reflects the complex anatomy of the mandible and the frequent need for invasive surgical techniques, which expose patients to greater risks of infection. The mandible’s proximity to the oral cavity, which harbors a rich microbiota, increases the risk of bacterial contamination during both injury and surgical repair. Additionally, the mandible endures significant functional stress during mastication, which can compromise the stability of fixation devices and impede healing.

In contrast, lower infection rates were observed in condylar (1.8%), midface (0.9%), and frontal bone fractures (2.7%). These regions benefit from less invasive surgical techniques, such as closed reduction and extraoral approaches, and possess unique anatomical characteristics that may contribute to better postoperative outcomes regarding infection prevention. The midface and frontal bones have a richer vascular network compared to the mandible, potentially enhancing immune response and tissue repair. These findings emphasize the importance of tailored surgical approaches based on fracture location and suggest that less invasive methods may reduce infection risks in specific anatomical regions. Despite these findings, significant heterogeneity was observed across the included studies, as reflected by an I^2^ statistic of 92.6%. This heterogeneity likely stems from variations in study designs, population characteristics, surgical techniques, antibiotic protocols, and definitions of FRIs. Importantly, our analysis revealed no significant differences in infection rates over time, suggesting that advances in surgical techniques or perioperative care have not markedly reduced FRI incidence. Similarly, no differences were observed between studies reporting prophylactic antibiotic use and those that did not, raising questions about the optimal role of antibiotics in FRI prevention.

The wide variation in infection rates by anatomical site calls for targeted preventive strategies, especially for mandibular fractures in which the risk is highest. Rigorous perioperative management, including meticulous surgical technique and careful patient selection, is essential. Delayed surgical intervention was associated with significantly higher infection rates. Malanchuk and Kopchak observed infection rates escalating from 4.2% when treated within 24 h to 55% when treatment was delayed beyond seven days [7]. Early surgical intervention reduces the time for bacterial colonization and biofilm formation at the fracture site, thereby decreasing the likelihood of infection.

Patient-specific factors play a crucial role in the development of FRIs. Smoking emerged as a consistent risk factor, with studies demonstrating a higher incidence of postoperative infections among smokers. Agnihotri et al. reported a 30% higher complication rate in smokers compared to non-smokers undergoing mandibular fracture repair [24]. Smoking impairs wound healing through vasoconstriction, reduced immune function, and decreased collagen synthesis [33]. The consistent association between smoking and infection emphasizes the need for perioperative smoking cessation programs to mitigate these risks.

Substance abuse, particularly intravenous drug use, also significantly increased infection rates due to immunosuppression and poor compliance with postoperative care. Preoperative screening and intervention for substance abuse may help reduce postoperative complications by addressing these underlying issues.

Similarly, advanced age and comorbid conditions such as diabetes mellitus were consistently associated with higher infection rates. Elderly patients often have reduced bone density, impaired immune responses, and multiple comorbidities that compromise healing [34]. Diabetic patients exhibit impaired neutrophil function and collagen synthesis, leading to delayed wound healing and increased susceptibility to infection [35]. Strict glycemic control and the optimization of comorbid conditions preoperatively are critical to improving outcomes in these populations.

The choice of surgical technique and fixation method significantly influences infection rates. Minimally invasive approaches that preserve soft tissue integrity and vascular supply have been associated with lower infection rates [36]. For instance, Li et al. (2016) reported a 5% infection rate using the supratemporalis approach for intracapsular condylar fractures, compared to 15% with conventional methods [37].

Prophylactic antibiotic use is a cornerstone of infection prevention in maxillofacial surgery. Standardized protocols involving a single preoperative dose followed by 24–48 h of postoperative antibiotics have been shown to be effective. Andreasen et al. demonstrated that antibiotic use reduced infection rates to 7% compared to no use, and a single administered dose might be equally effective or better compared to a prolonged course [8]. However, variability in antibiotic selection and duration across studies suggests a need for consensus guidelines to optimize prophylactic strategies and minimize the development of antibiotic resistance.

Preoperative oral and periodontal health significantly impacts postoperative infection rates. Poor oral hygiene and the presence of periodontal disease increase the bacterial load in the oral cavity, elevating the risk of surgical site contamination [14]. In another study, the authors found a significant association between the severity of periodontal disease and the incidence of surgical site infections [38]. Patients with severe periodontitis were found to be over seven times more likely to develop infections compared to those without periodontal disease. This highlights the importance of periodontal health in fracture management, suggesting that severe periodontitis significantly increases the risk of postoperative infections, whereas smoking, often considered a confounding factor, was not statistically significant in this context. Preoperative dental assessments and treatments can reduce bacterial colonization, thereby decreasing infection risks. These findings emphasize the importance of promoting oral health as part of comprehensive perioperative management.

This systematic review has several limitations that should be acknowledged. Firstly, the heterogeneity in study designs, populations, and definitions of FRIs across the included studies may have impacted the comparability and generalizability of the findings. Significant variability in surgical techniques, antibiotic regimens, and patient demographics, as well as inconsistent risk factor reporting, could introduce biases, limiting the reliability of the pooled infection rates and identified risk factors. A detailed analysis of infection rates based on specific mandibular fracture sites was not feasible, as many included studies did not distinguish between alveolar and non-alveolar involvement or other anatomical subcategories. To maintain a comprehensive and consistent overview, we opted not to make this distinction in our meta-analysis. Additionally, the reliance on observational studies and the lack of high-quality randomized controlled trials (RCTs) limit the strength of evidence for the management strategies proposed. Many studies also lacked control groups or sufficient follow-up periods, further affecting the robustness of outcomes related to FRI risk factors and intervention efficacy. Future research should aim to incorporate a more comprehensive assessment of comorbidities to better understand their impact on infection risk and treatment outcomes. Lastly, publication bias cannot be ruled out, as studies reporting null findings on FRIs may be underrepresented, potentially skewing the reported incidence and effectiveness of certain preventive strategies. Future research with standardized methodologies, consistent definitions, and a focus on high-quality RCTs is necessary to strengthen the evidence base for FRI management in maxillofacial trauma.

The effective management of fracture-related infections (FRIs) in maxillofacial trauma requires an individualized approach that considers the type of fracture, infection severity, and patient-specific risk factors. Broadly, management can be categorized into nonoperative and operative approaches. Nonoperative measures, such as antibiotic therapy tailored to culture results and local wound care, are effective for early-stage infections or superficial plate exposure without signs of deep infection. In cases in which operative intervention is necessary, techniques such as debridement, removal of infected hardware, and stabilization using external fixation or rigid internal fixation have been employed [32,39,40]. In our meta-analysis, the timing of fracture-related infections (FRIs) following surgical treatment was not consistently reported across the included studies. As a result, we were unable to assess whether infection rates varied based on the postoperative timeframe. This represents a significant gap in the current literature, as understanding when FRIs most commonly occur could help refine antibiotic prophylaxis strategies and postoperative monitoring protocols. Future studies should aim to capture this information to improve infection prevention and management in mandibular fractures. Studies included in this review do emphasize the importance of early surgical debridement to reduce bacterial load and facilitate proper healing. Additionally, ensuring that the patient adheres to optimal postoperative care, including oral hygiene maintenance and follow-up visits, significantly contributes to reducing recurrence rates.

## 5. Conclusions

In conclusion, the infection rates observed in this review underscore the critical role of both surgical technique and patient-specific factors in influencing postoperative outcomes. The higher infection rate in mandibular fractures reflects the need for targeted preventive strategies and tailored surgical approaches. Addressing modifiable risk factors such as periodontal health, smoking, and substance abuse, optimizing the management of comorbid conditions, and implementing standardized antibiotic protocols are essential strategies for reducing infection rates. Furthermore, the results of this systematic review indicate that a multidisciplinary approach, including collaboration between surgeons, infectious disease specialists, and primary care providers, might be the best way forward to enhance the outcomes for patients with FRIs.

Future research should focus on establishing standardized guidelines, conducting high-quality randomized controlled trials, and exploring innovative therapies to enhance patient outcomes.

## Figures and Tables

**Figure 1 jcm-14-01332-f001:**
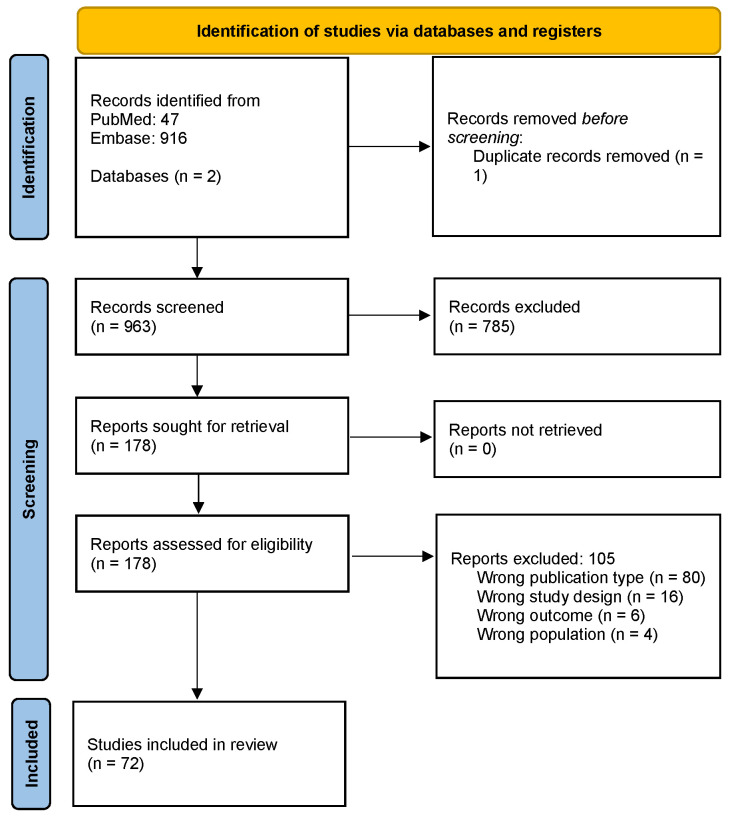
PRISMA Flowchart.

**Figure 2 jcm-14-01332-f002:**
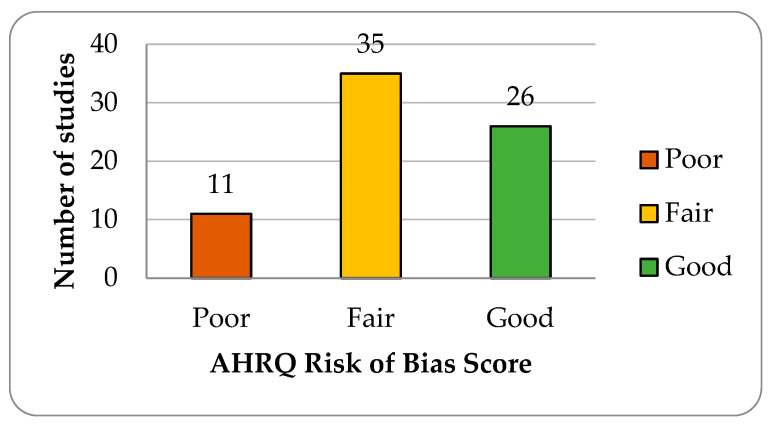
Distribution of studies by Agency for Healthcare Research and Quality (AHRQ) Risk of Bias score.

**Figure 3 jcm-14-01332-f003:**
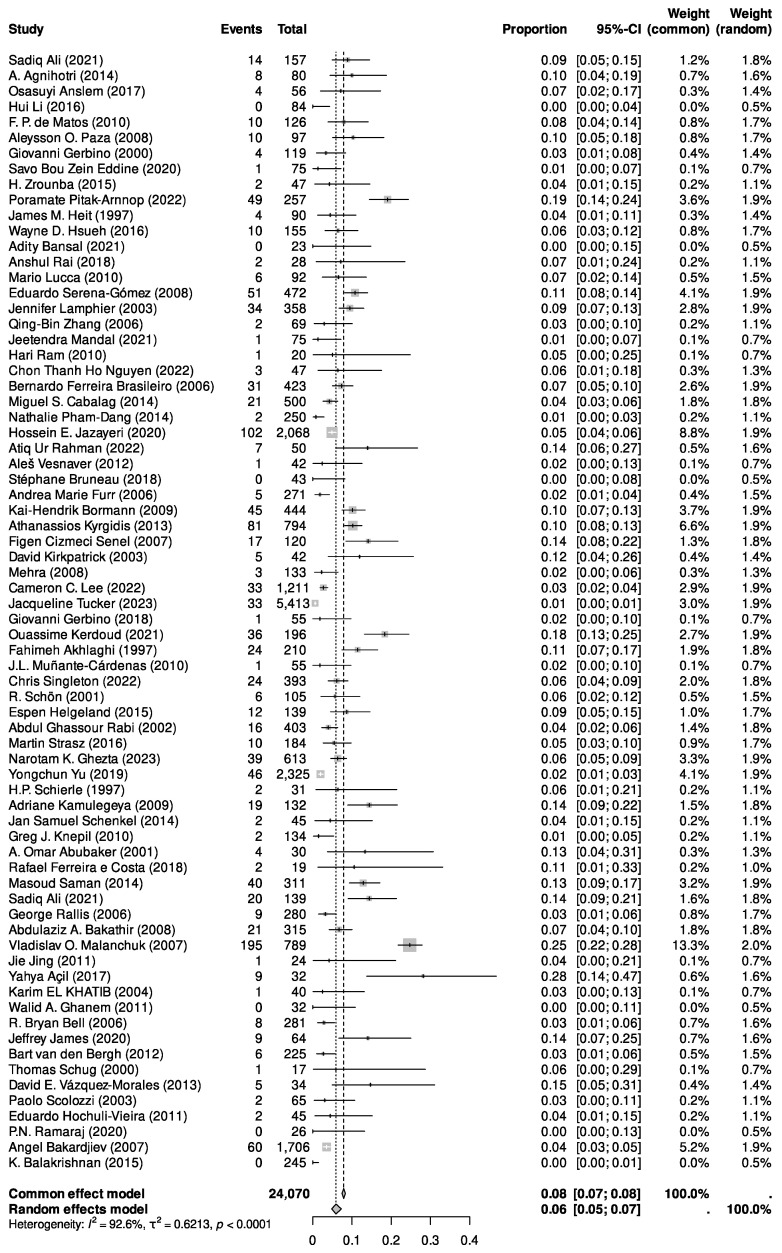
A forest plot of a single-arm meta-analysis showing study proportions with 95% CIs, pooled estimates (fixed and random effects), and study weights.

**Figure 4 jcm-14-01332-f004:**
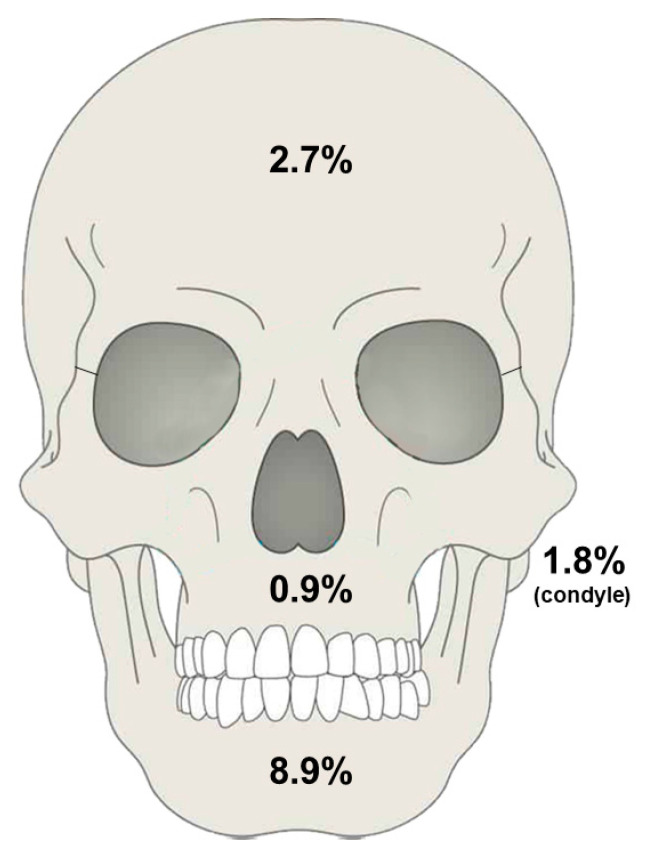
Pooled infection rates across the mandible, condyle, midface, and frontal bone area.

**Figure 5 jcm-14-01332-f005:**
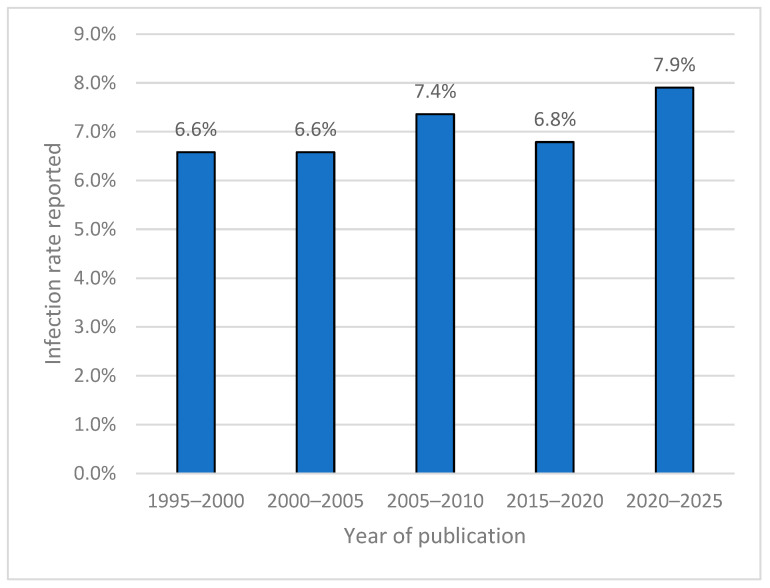
Reported infection rates when stratifying by publication year.

**Table 1 jcm-14-01332-t001:** Search strategy.

Medline
(“Surgery, Oral” [MeSH Terms] OR “oral surgery” [Text Word] OR “maxillofacial surgery” [Text Word]) AND (“Fractures, Bone” [MeSH Terms] OR “fracture” [Text Word]) AND (“Surgical Wound Infection” [MeSH Terms] OR “osteomyelitis” [Text Word] OR “fracture-related infection” [Text Word])
Embase
(“oral surgery”/exp OR “maxillofacial surgery”/exp OR “oral surgery”:ab,ti OR “maxillofacial surgery”:ab,ti) AND (“bone fracture”/exp OR “fracture”:ab,ti) AND (“surgical wound infection”/exp OR “infection”/exp OR “osteomyelitis”/exp OR “infection”:ab,ti OR “osteomyelitis”:ab,ti OR “fracture related infection”:ab,ti)

**Table 2 jcm-14-01332-t002:** Categories of fracture-related infections in oral and maxillofacial surgery.

Stage	Definition	Clinical Features
**1**	Plate or fracture exposure without clinical signs	Fracture or hardware exposure in the absence of pain, swelling, or purulent discharge.
**2**	Plate or fracture exposure with infectious signs	Fracture or hardware exposure accompanied by signs of infection, including pain, redness, some swelling, and spontaneous drainage of pus.
**3**	Fracture-related abscess	Localized collection of pus near fracture site or hardware. Pain, tenderness, fluctuation, and erythema around affected area.
**4**	Fracture-related, multi-space infection	Pain, swelling, fluctuation, and clinical or radiographic evidence of multi-space infection from the fracture site to the adjacent fascial spaces.
**5**	Fracture-related osteomyelitis	Infection involving the bone including pain, swelling, fluctuation, intra- and/or extraoral redness, pus drainage, and radiographic evidence of bone destruction, often accompanied by an unstable fracture or failing hardware (nonunion).

## Data Availability

The original contributions presented in this study are included in the article/Appendix A. Further inquiries can be directed to the corresponding author.

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
