# Peer review of "A Systematic Review and Meta-Analysis of Fracture-Related Infections in Maxillofacial Trauma: Incidence, Risk Factors, and Management Strategies"

_jcm, 2025, doi:10.3390/jcm14041332_

Round 1
Reviewer 1 Report
Comments and Suggestions for Authors
Dear Authors
The authors presented an interesting review of the literature on FRI fracture-related infections. The work is based on 40 correctly selected citations. The topic is interesting, and the presented results have been rightly grouped, taking into account various etiological factors.
However, I have a few comments:
1. Keywords should be in alphabetical order
2. In Results: FRI analysis of mandibular fractures should be detailed into fractures of the body ( with and without teeth), fractures of the angle, ramus, and condylar process (this information is included in the text). Body fractures in the dentition section will certainly have a different FRI than the other locations.
3. It is interesting that only four studies included data on FRIs and antibiotic use. Please specify what types of antibiotics were used. The text contains the following information: „In a comparative analysis of ceftriaxone versus penicillin for antibiotic prophylaxis of compound mandibular fractures, both regimens showed similar efficacy with only two infections reported per group, which resolved with local 286 wound care, removal of the internal fixation, and oral antibiotics”. Did all eligible studies use only these two antibiotics?
4. I also did not find information on how long after the surgical treatment FRIs occurred, which may also be interesting information for analysis
5. There is also a lack of information on what other comorbidities (apart from the diabetes mentioned) were considered in the analysis.
In summary: The article is very interesting, and after detailing a few issues, it will certainly be a valuable source of information on Fracture-related infections (FRIs)
Author Response
Comment 1: Keywords should be in alphabetical order
Response 1: we changed this to alphabetical
Comment 2: In Results: FRI analysis of mandibular fractures should be detailed into fractures of the body (with and without teeth), fractures of the angle, ramus, and condylar process (this information is included in the text). Body fractures in the dentition section will certainly have a different FRI than the other locations.
Response 2: Thank you for your insightful comment. We acknowledge that within the non-condylar mandible, the infection rate (FRI) will vary depending on the specific anatomical site involved. However, our meta-analysis did not allow for a detailed analysis of the different mandibular fracture sites. As you rightfully noted, the FRI may differ based on whether fractures involve the alveolar or non-alveolar regions. Unfortunately, many of the included studies did not distinguish between these subcategories or provide sufficient data to enable a granular analysis. To ensure a comprehensive and consistent overview, we decided not to differentiate between these fracture types in our analysis. We have clarified this limitation in the manuscript accordingly.
Comment 3: It is interesting that only four studies included data on FRIs and antibiotic use. Please specify what types of antibiotics were used. The text contains the following information: „In a comparative analysis of ceftriaxone versus penicillin for antibiotic prophylaxis of compound mandibular fractures, both regimens showed similar efficacy with only two infections reported per group, which resolved with local 286 wound care, removal of the internal fixation, and oral antibiotics”. Did all eligible studies use only these two antibiotics?
Response 3: We appreciate the reviewer’s interest in the antibiotic regimens included in our meta-analysis. As noted, only four studies provided detailed data on fracture-related infections (FRIs) and the effect of antibiotic use. The antibiotics reported in these studies varied, with some studies specifying the use of first-generation amoxicillin-clavulanic acid, metronidazole, and macrolides. However, not all studies explicitly distinguished between specific antibiotic regimens, and as such, our analysis was limited in its ability to compare efficacy across different antibiotic types. A more detailed breakdown of antibiotic usage in each study has now been added to the manuscript.
Comment 4: I also did not find information on how long after the surgical treatment FRIs occurred, which may also be interesting information for analysis
Response 4: Unfortunately, this information was not reported in the studies included in our meta-analysis. As a result, we were unable to analyze the timing of FRIs in relation to surgical intervention. Recognizing the clinical importance of this factor, we have highlighted this gap in our discussion as a future opportunity for research in this field. A more detailed evaluation of the temporal occurrence of FRIs could provide valuable insights into optimal post-surgical management and prophylactic strategies.
Comment 5: There is also a lack of information on what other comorbidities (apart from the diabetes mentioned) were considered in the analysis.
Response 5: No specific considerations were made when selecting papers based on comorbidities, as our inclusion criteria did not exclude studies based on patient health conditions. Additionally, there was a lack of detailed information on comorbidities beyond diabetes, HIV, COVID, substance abuse and periodontal disease, which was the only condition explicitly mentioned in some studies. Future research should aim to incorporate a more comprehensive assessment of comorbidities to better understand their impact on infection risk and treatment outcomes. This was added to our discussion paragraph.
Reviewer 2 Report
Comments and Suggestions for Authors
as you will be able to see from the review I don't have many questions to ask, because the article is well structured in its parts. The purpose of the paper is clear, the materials and methods are easy to understand and repeatable, the results clearly stated, and the discussion emphasizes the limitations of the present study.
I have only one question for the authors:
Was gender and age difference considered in the distribution of fracture risk rates?
Author Response
Comment 1: Was gender and age difference considered in the distribution of fracture risk rates?
Response 1: Thank you for willing to review our paper. Gender and age details are included in our Supplemental Table 2. However, most of the included studies did not provide stratified data on age and gender in relation to fracture-related infection (FRI) rates. Due to this limitation, it was not feasible to incorporate a separate risk factor analysis for age and gender in our meta-analysis. Future studies with more granular reporting on these variables would allow for a more detailed assessment of their potential impact on FRI risk.